# Towards a Unified Conversational Recommendation System: Multi-task Learning via Contextualized Knowledge Distillation

**Yeongseo Jung    Eunseo Jung    Lei Chen**
The Hong Kong University of Science and Technology
{yjungag, ejungab, leichen}@cse.ust.hk

## Abstract

In Conversational Recommendation System (CRS), an agent is asked to recommend a set of items to users within natural language conversations. To address the need for both conversational capability and personalized recommendations, prior works have utilized separate recommendation and dialogue modules. However, such approach inevitably results in a discrepancy between recommendation results and generated responses. To bridge the gap, we propose a multi-task learning for a unified CRS, where a single model jointly learns both tasks via **Con**textualized **K**nowledge **D**istillation (ConKD). We introduce two versions of ConKD: *hard gate* and *soft gate*. The former selectively gates between two task-specific teachers, while the latter integrates knowledge from both teachers. Our gates are computed on-the-fly in a context-specific manner, facilitating flexible integration of relevant knowledge. Extensive experiments demonstrate that our single model significantly improves recommendation performance while enhancing fluency, and achieves comparable results in terms of diversity.[1]

## 1 Introduction

Natural language dialogue systems generally fall into either task-oriented system (Wen et al., 2016; Henderson et al., 2019; Peng et al., 2020) or open-domain dialogue system (Xing et al., 2016; Zhang et al., 2020; Adiwardana et al., 2020). Despite the same modality (conversation), the tasks differ in their objectives; the former aims to achieve certain tasks (e.g. booking hotels), while the latter engage in an open-ended dialogue.

Conversational Recommendation (CR) is an emerging task in natural language dialogue, which combines the task-oriented and open-domain (Gao et al., 2021). The task aims to recommend proper

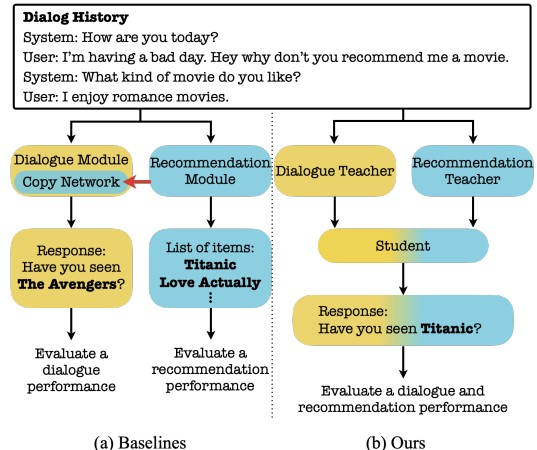

Figure 1: Illustration of evaluation mismatch. Previous methods evaluate recommendation and dialogue performance independently as separate models are trained for each task. In contrast, a single model is utilized for both tasks in ConKD.

items to users through natural language conversations, and the model-generated conversation is expected to be fluent and suit the context. Unlike traditional recommendation systems, the interactive nature of multi-turn dialogue allows the agent to explore explicit user interests that may not be present in the user's history. This advantage particularly stands out compared to the traditional models that have access to only few purchase histories or implicit feedback (e.g. click) from users.

To generate appropriate responses containing recommended items aligned with a user's taste, an agent needs to possess both recommendation and dialogue capabilities. Previous works addressed the issue with separate modules (Li et al., 2018; Chen et al., 2019; Zhou et al., 2020; Lu et al., 2021). A recommendation module learns user preferences based on conversation history and retrieve relevant items, while a dialogue module generates the final response sentences. In Conversational Recommendation System (CRS), a major problem lies in incorporating the two separate modules. Common

---

[1]The code is available at https://github.com/yeongseoj/ConKD

strategies for injecting recommendation ability into the responses include copy mechanism (Gu et al., 2016) and pointer network (Gulcehre et al., 2016).

Despite these efforts, the prior approaches (Li et al., 2018; Chen et al., 2019; Zhou et al., 2020; Lu et al., 2021) have demonstrated a discrepancy between the separate modules: *the results of a recommendation model are not integrated in the generated response* as depicted in Figure 1. The dialogue module suggests "The Avengers", while the recommendation module recommends "Titanic", revealing a clear disagreement between the two modules. Accordingly, the probability distribution of the recommendation model is not directly reflected in the output of the dialogue model. Such mismatch is inevitable when CR is formulated as two separate tasks, failing to serve the original purpose of the task.

To address this challenge, we propose a multi-task learning approach for a unified CRS using **Con**textualized **K**nowledge **D**istillation (ConKD). We build a CRS with a single model via knowledge transfer by two teacher models, a dialogue teacher and a recommendation teacher. However, combining the knowledge is not straightforward due to the nature of CRS; the task differs in each time step depending on the context.

In this light, we introduce two gating mechanisms, *hard gate* and *soft gate*, to effectively fuse teachers' knowledge. With hard gate, knowledge transfer comes solely from either of the teachers, while soft gate integrates both sources of knowledge. We introduce an adaptive nature in the gates which is **context-specific** and computed **on-the-fly** during forward propagation. To our knowledge, this is the first work to explicitly demonstrate the existence of the discrepancy, and provide a dedicated training mechanism to address it. Moreover, our approach offers the flexibility in selecting model architectures, enabling integration of diverse language and recommendation models.

Extensive experiments conducted on a widely used benchmark dataset (Li et al., 2018) demonstrate that our single model significantly outperforms baselines in terms of recommendation performance and response fluency, while achieving comparable results in response diversity.

The contributions of our work can be summarized as follows:

- We propose a multi-task learning approach for a unified CRS using **Con**textualized **K**nowledge

**D**istillation.

- We introduce two versions of ConKD, employing different gate mechanisms: *hard gate* and *soft gate*.

- Our approach surpasses strong baseline models in making coherent recommendations and fluent responses, while competitive results are observed in response diversity.

## 2 Preliminaries & Related Work

### 2.1 Open-Ended Conversational Recommendation System

Conventional recommendation systems mainly focus on building a static user preference based on previous histories, such as clicks, purchases, and ratings (Sarwar et al., 2001; Koren et al., 2009; Kuchaiev and Ginsburg, 2017; Kang and McAuley, 2018). In such environment, where feedback from users is static, implicit, and sparse, recommendation systems have difficulty reflecting dynamic changes in users' preferences as well as suffer the cold-start problem (Gao et al., 2021).

ReDial (Li et al., 2018) is one of the first attempts at handling such issue; the work combines open-ended chit-chat with recommendation task, called Conversational Recommendation System (CRS). Specifically, let $(\mathbf{x}, \mathbf{y})$ be a dialogue sample, where $\mathbf{x} = \{x^1, x^2, ..., x^m\}$ is a set of previous dialogue turns. $m$ is the lengths of turns in the dialogue history and $\mathbf{y}$ is the corresponding response (ground truth). In each turn, a recommendation module is expected to provide an item set $I_u$ for a user $u \in U$, while a dialogue module produces a response $\mathbf{y}$ based on a dialogue history $\mathbf{x}$.

To incorporate recommended items into a response, a copy mechanism (Gu et al., 2016) or pointer network (Gulcehre et al., 2016) is generally adopted in prior works (Li et al., 2018; Chen et al., 2019; Zhou et al., 2020; Lu et al., 2021; Zhou et al., 2023). In such methods, additional networks are trained to predict whether the next token is a word or an item by aggregating representations from recommendation and dialogue modules.

In the aim of improving a CRS, previous studies leverage external knowledge in training. In KBRD (Chen et al., 2019), an item-oriented knowledge graph is introduced as an auxiliary input to a recommendation module. KGSF (Zhou et al., 2020) utilizes both word-level and item-level knowledge graphs in training a CRS. In addition to the knowl-

| Models | Mismatch | R@1 | ReR@1 | R@10 | ReR@10 | R@50 | ReR@50 |
|--------|----------|-----|-------|------|--------|------|--------|
| KBRD | 0.931 | 0.034 | 0.008 (76.47%) | 0.168 | 0.040 (76.19 %) | 0.360 | 0.096 (74.17%) |
| KGSF | 0.926 | 0.038 | 0.008 (78.95%) | 0.183 | 0.043 (76.50%) | 0.382 | 0.109 (73.33%) |
| RevCore | 0.971 | 0.052 | 0.006 (88.46%) | 0.195 | 0.031 (84.10%) | 0.341 | 0.077 (77.42%) |

Table 1: Comparisons of Recall (**R@**$k$) by recommendation modules and Recall in Response (**ReR@**$k$) by dialogue modules. Numbers in parenthesis indicate relative decrease in recall score by dialogue module compared to the recall score by a recommendation module. The scores are averaged over three runs with random seeds.

edge graphs, movie review data is utilized in RevCore (Lu et al., 2021), and the meta information of items is encoded in (Yang et al., 2022) to enrich item representation. However, these works employ separate modules to manage CRS, which inevitably introduces discrepancy issues.

To address this problem, prompt-based learning strategies are introduced in (Wang et al., 2022b; Deng et al., 2023). Despite the unified architecture, these approaches fail to dynamically incorporate multiple recommendations into a response. RecInDial (Wang et al., 2022a) introduces a vocabulary pointer and knowledge bias to produce a unified output by combining two modules.

## 2.2 Knowledge Distillation

The core idea behind Knowledge Distillation (KD) (Hinton et al., 2015) is transferring knowledge of a high-capacity teacher network to a relatively smaller student model. In knowledge distillation, a student network is guided by not only a one-hot encoded ground-truth but also a soft target mapped by the teacher network (probability distribution). This is known to transfer a class-wise relation mapped by a teacher is commonly termed the *dark knowledge*. Given a data sample from a joint distribution $(x, y) \in \mathcal{X} \times \mathcal{Y}$, a student model is optimized by combining two cross-entropy terms.

$$\mathcal{L}_{\text{KD}}(\theta) = -\sum_{k=1}^{|Y|} \gamma \hat{y}_k \log P_\theta(y_k|x)$$
$$+ (1 - \gamma) \tilde{P}_\phi(y_k|x) \log \tilde{P}_\theta(y_k|x) \quad (1)$$

where $|Y|$ and $\hat{y}_k$ denote the number of classes and a $k$-th target label (one-hot encoded) respectively. $\gamma$, and $\tilde{P}$ denote a balancing parameter, and a probability distribution scaled with a temperature. $\theta$ and $\phi$ are parameters of a student and teacher network respectively.

## 3 Unified CRS via ConKD

In this section, we first demonstrate the mismatch issue with preliminary experiments and introduce our approach that mitigates such problem.

### 3.1 Preliminary Experiments

In our preliminary experiments on REDIAL (Li et al., 2018) dataset[2], we aim to identify the mismatch problem in evaluation. In Table 1, we compare the recommendation results from two separate modules using recall metrics: R@$k$ (Recall) and ReR@$k$ (Recall in Response) which evaluates the top-$k$ items predicted by a recommendation module and a dialogue module respectively. In all metrics, a significant degradation is observed when recall is computed on the dialogue response. The relative decreases in the performance are ranged from 73.33% to 88.46%, implying that *a large discrepancy* exists between the outputs of recommendation modules and generated responses. However, incorporating the recommendation module's outputs during inference does not provide the fundamental solution to the problem. Further discussion on this issue is provided in Appendix D.

To address the discrepancy, we propose a multitask learning approach for a unified conversational recommendation system via **Con**textualized **K**nowledge **D**istillation (ConKD). ConKD consists of three key components: a dialogue teacher and a recommendation teacher as experts on each task, and a student model - a multi-task learner, as described in Figure 2.

### 3.2 Recommendation Teacher

A recommendation teacher models the item-user joint distribution and provides a set of items that suit a user's preference. We adopt the model structure of (Zhou et al., 2020), where an item-oriented Knowledge Graph (KG) (Bizer et al., 2009) and word-oriented KG (Speer et al., 2016) are encoded to build a user preference. To learn item representations with structural and relational information, R-GCN (Schlichtkrull et al., 2017) is adopted for

---

[2]Li et al. (2018) proposed a dataset and a model, which we term the dataset as REDIAL and the model as ReDial hereinafter.

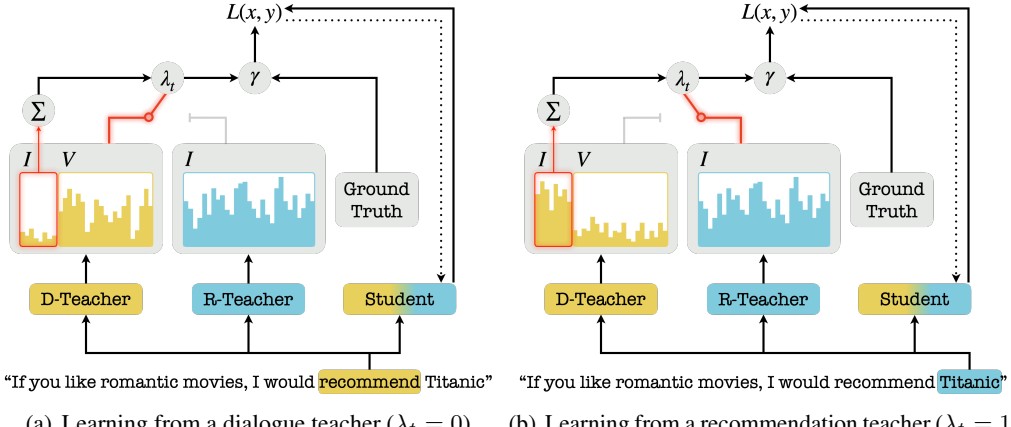

(a) Learning from a dialogue teacher ($\lambda_t = 0$)  (b) Learning from a recommendation teacher ($\lambda_t = 1$)

Figure 2: The main structure of the proposed contextualized knowledge distillation with the hard gate. D-Teacher and R-Teacher denote dialogue teacher and recommendation teacher respectively. $I$ and $V$ denote item space and vocabulary space respectively. The dashed arrow indicates backward propagation. One can easily extend the above to the soft gate, where $\lambda_t$ is continuous.

the item-oriented KG as follows.

$$\mathbf{h}_e^{(l+1)} = \sigma(\sum_{r \in \mathcal{R}} \sum_{e' \in \mathcal{E}_e^r} \frac{1}{Z_{e,r}} \mathbf{W}_r^{(l)} \mathbf{h}_{e'}^{(l)} + \mathbf{W}^{(l)} \mathbf{h}_e^{(l)})$$
(2)

where $\mathbf{h}_e^{(l)}$ denotes the representation of node $e$ at $(l)$-th layer and $\mathbf{h}^{(0)}$ is the initial node embedding. $\mathcal{E}_e^r$ is the set of neighbor nodes for node $e$ under the relation $r$. $\mathbf{W}_r^{(l)}$ and $\mathbf{W}^{(l)}$ are learnable matrix for handling various edge types and self-loop respectively. $Z_{e,r}$ is a normalization constant. Similarly, word-oriented KG is encoded with the GCN (Kipf and Welling, 2016) and the description in detail is illustrated in Appendix B.

Given the learned node embeddings, a user representation $\mathbf{p_u}$ is acquired by aggregating words $\mathbf{v}^{(\mathbf{x})}$ and items $\mathbf{n}^{(\mathbf{x})}$ that appear in previous dialogue turns $\mathbf{x}$ as follows[3].

$$\mathbf{p}_u = \beta \cdot \mathbf{v}^{(\mathbf{x})} + (1 - \beta) \cdot \mathbf{n}^{(\mathbf{x})}$$
$$\beta = \sigma(\mathbf{W_g}[\mathbf{v}^{(\mathbf{x})}; \mathbf{n}^{(\mathbf{x})}])$$
(3)

where $\mathbf{W_g}$ are learnable parameters for computing the balancing parameter $\beta$. Finally, a matching score between a user $u$ and an item $i$ is calculated as follows.

$$P_\psi(i) = \mathsf{softmax}(\mathbf{p}_u^T \mathbf{n}_i)$$
(4)

where $\psi$ is model parameters optimized to maximize the likelihood of predicting ground-truth items.

---
[3]For the detailed aggregation process, please refer to Section 4.3 in (Zhou et al., 2020)

### 3.3 Dialogue Teacher

To handle the chit-chat task, we train a conditional language model that intakes dialogue history and generates a context-aware response. We explore two primary structural variations for our language model:

- **KGSF** (Zhou et al., 2020): A standard transformer (Vaswani et al., 2017) and a knowledge-enhanced transformer (Zhou et al., 2020) are utilized for each as an encoder and a decoder.

- **DialoGPT** (Zhang et al., 2020): A transformer-based pre-trained language model (PLM) trained on a large-scale dialogue dataset.

Both dialogue models are trained to maximize the likelihood of predicting the ground truth response as follows:

$$\mathcal{L}(\phi) = -\sum_{j=1}^{|Y|} \sum_{t=1}^{T} \frac{1}{T} \hat{y}_{t,j} \log P_\phi(y_{t,j} | \mathbf{y}_{1:t-1}, \mathbf{x})$$
(5)

where $T$ and $j$ are the length of a response $\mathbf{y}$ and a token index respectively. $Y$ is a union of the vocabulary set and item set ($Y = V \cup I$), and $\phi$ denotes the parameters of the dialogue model.

### 3.4 Contextualized Knowledge Distillation

We elaborate on how a student model learns both the recommendation and dialogue tasks. Specifically, two gating mechanisms are introduced: *discrete* and *continuous* gate, which integrate knowledge between the two teachers in an adaptive manner.

## Hard Gate

A student model is trained to minimize the gap between its conditional probability distribution and the conditional probabilities mapped by the teacher networks. However, *it is not ideal to equally weight the knowledge from both teacher models at every time step* as seen in the following response.

> *y1: If you like romance movies, I would recommend **Titanic**.*

At the last time step, where the item *Titanic* appears, knowledge of a recommendation teacher can help a student learn the recommendation ability. On the contrary, a student can learn to generate a coherent and suitable utterance by accessing knowledge from the dialogue model except the time of recommendation.

Taking all these factors into account, we introduce a token-level hard gate between teacher networks, where supervision solely comes from either of the teachers at each time step. To distinguish which knowledge of the two teachers to be transferred, we *aggregate the probability mass of item indexes mapped by the dialogue teacher in each time step*. This is built on an assumption that the dialogue model assigns a relatively large probability mass on item indexes at the time of recommendation.

Given the distribution $P_\phi(y_t|\mathbf{y}_{1:t-1}, \mathbf{x})$ mapped by the dialogue teacher, we calculate a sum of item probabilities which answers the question of "*is this time to recommend an item?*". Therefore, a **time step-specific** hard gate is computed **on-the-fly** during forward propagation as follows.

$$\lambda_t = \begin{cases} 0, & \text{if } \sum_{y' \in \mathcal{I}} P_\phi(y'|\mathbf{y}_{1:t-1}, \mathbf{x}) < \eta \\ 1, & \text{otherwise} \end{cases} \quad (6)$$

where $\mathcal{I}$ denotes the set of items in the vocabulary, and $\eta$ is a predefined threshold. When $\lambda_t$ is computed to be 1, it is a clear indication that a CRS is expected to output a recommendation result. On the contrary, when the hard gate is 0, a dialogue teacher defines the current time step as a dialogue time step; hence, the CRS is expected to make a coherent chit-chat.

## Soft Gate

The hard gate inevitably introduces a hyper-parameter $\eta$ due to the thresholding approach. To remove the hyper-parameter search on $\eta$, we introduce a continuous gating mechanism. This can be applied under the assumption that a sum of item probabilities mapped by the dialogue teacher reflects the extent to which recommendation is expected. Therefore, the aggregated mass answers the question of '*how much to learn the recommendation ability at the time step*". Based on the intuition, we introduce a soft gate as follows.

$$\lambda_t = \sum_{y' \in \mathcal{I}} P_\phi(y'|\mathbf{y}_{1:t-1}, \mathbf{x}) \quad (7)$$

where the gate $\lambda_t$ takes continuous values within the range of $[0, 1]$. A gate close to 1 indicates that the system should focus more on recommendation, while a gate close to 0 suggests that the agent is expected to prioritize the conversation task.

To validate our assumption regarding the behavior of the dialogue teacher, we conducted a preliminary experiment using a smaller model, KGSF. We computed the average sum of item probabilities in a dialogue time step $\lambda_v$ and in a recommendation time step $\lambda_r$. The computed value of $\lambda_r$ was found to be $0.653$, while $\lambda_v$ was measured to be $0.023$. These results support our assumption that *the dialogue teacher assigns relatively large probability mass to item indexes in recommendation time*. We provide further discussion on the validity of $\lambda$ in Appendix C.

The gating computation differs from the previous gating approaches (Zhou et al., 2020; Li et al., 2018; Chen et al., 2019; Lu et al., 2021) in two ways : 1) we leverage a teacher distribution as a signal of gating, where the gates can be discrete $\lambda_t \in \{0, 1\}$ or continuous $\lambda_t \in [0, 1]$. 2) our gates are not learned but a simple sum of probabilities.

## Contextualized Knowledge Distillation Loss

With the two pre-trained teachers and the gating mechanisms, we now introduce loss for contextualized knowledge distillation.

KD losses for each task are formulated as follows.

$$\mathcal{L}_{\text{DIAL-KD}}(\theta) = -\sum_{k=1}^{|Y|} \sum_{t=1}^{T} \frac{1}{T} P_\phi(y_{t,k}|\mathbf{y}_{1:t-1}, \mathbf{x}) \times$$
$$\log P_\theta(y_{t,k}|\mathbf{y}_{1:t-1}, \mathbf{x}) \quad (8)$$

$$\mathcal{L}_{\text{REC-KD}}(\theta) = -\sum_{k=1}^{|Y|} \sum_{t=1}^{T} \frac{1}{T} P_\psi(y_{t,k}|\mathbf{x}) \times \quad (9)$$
$$\log P_\theta(y_{t,k}|\mathbf{y}_{1:t-1}, \mathbf{x})$$

where $\theta$ is the parameter of the student. Then, the final KD losses for each task are derived as follows.

$$\mathcal{L}_{\text{DIAL}}(\theta) = (1 - \gamma)\mathcal{L}_{\text{NLL}} + \gamma\mathcal{L}_{\text{DIAL-KD}}$$
$$\mathcal{L}_{\text{REC}}(\theta) = (1 - \gamma)\mathcal{L}_{\text{NLL}} + \gamma\mathcal{L}_{\text{REC-KD}} \quad (10)$$

where $\gamma$ is the balancing parameter between ground truth and teacher distribution, and $\mathcal{L}_{\text{NLL}}$ is the cross entropy with ground truth. Finally, the losses are aggregated with our gate $\lambda_t$ per time step.

$$\mathcal{L}(\theta) = (1 - \lambda_t)\mathcal{L}_{\text{DIAL}} + \lambda_t\mathcal{L}_{\text{REC}} \quad (11)$$

When the hard gate is applied, a supervision is made by either of the teachers with the discrete $\lambda_t$. On the other hand, the soft gate fuses knowledge from the two teachers with the $\lambda_t$ being the weight. By optimizing the combined objective, a single model is capable of learning the dialogue and recommendation tasks simultaneously, alleviating the mismatch that comes from two separate modules in previous methods.

An evident advantage of employing contextualized knowledge distillation lies in taking the class-wise relation into consideration beyond the observed data (Hinton et al., 2015). With a one-hot encoded label, a neural network is trained to maximize the difference between the ground-truth and remaining classes; the dark knowledge is overlooked with one-hot supervision where a single index is set to 1 and the remaining indexes to 0s. In our work, the dark knowledge from both teachers is engaged in an adaptive manner to generate a fluent and user-specific response.

**Special Tokens**

Under CR, a dialogue turn falls into either a turn with recommendation result or a turn for querying a user preference. In this light, to inject an extra signal to the student model, we add two special tokens, [REC] and [GEN], at the beginning of each turn. *During training*, the ground truth prefix starts with either [REC] if the response includes an item , or [GEN] if it is chit-chat. This explicit scheme enables the model to learn turn-specific actions based on the preceding context and generate appropriate sequences.

*During inference*, we employ a pre-trained classifier to predict one of the two special tokens at each dialogue turn. The classifier is built with standard

transformer layers and optimized as follows.

$$\mathcal{L}(\theta_{cls}) = \mathbb{E}_{(k,x)\sim D}[-\sum_{j=1}^{|K|} \log P(k_j|\mathbf{x}; \theta_{cls})] \quad (12)$$

where $K = \{0, 1\}$ is the label set, and $\theta_{cls}$ denotes the classifier's parameters. The model learns to classify whether the current turn is intended for chit-chat or recommending an item, based on the dialogue history $\mathbf{x}$.

## 4 Experiments

### 4.1 Dataset

The proposed approach is tested on the recently introduced REDIAL (Recommendation Dialogues) dataset (Li et al., 2018). REDIAL is a conversation dataset which the dialogues are centered around recommendation, and the subject of the recommendation is movie. REDIAL contains 10,006 multi-turn dialogues, which amount to 182,150 utterances. The total number of unique movies in the dataset is 6,924, and the size of vocabulary is 23,928.

### 4.2 Baselines

**ReDial** (Li et al., 2018) consists of dialogue, recommendation, and sentiment analysis modules. Pointer network (Gulcehre et al., 2016) is introduced to bridge the modules. **KBRD** (Chen et al., 2019) introduces item-oriented KG (Bizer et al., 2009) and the KG representation is added when building a logit for the dialogue module (Michel and Neubig, 2018). **KGSF** (Zhou et al., 2020) integrates word-oriented KG (Speer et al., 2016) and item-oriented KG (Bizer et al., 2009) for semantic alignment. KG-enhanced transformer and copy network (Gu et al., 2016) are employed. **RevCore** (Lu et al., 2021) incorporates movie-review data for review-enriched item representations and utilizes a copy network (Gu et al., 2016). **DialoGPT** (Zhang et al., 2020) is fine-tuned on the REDIAL dataset. **RecInDial** (Wang et al., 2022a) finetunes DialoGPT-small with R-GCN(Schlichtkrull et al., 2017) and introduces a vocabulary pointer and knowledge-aware bias to generate unified outputs.

### 4.3 Evaluation Metrics

To validate the recommendation performance, we employ a top-$k$ evaluation approach with $k$ values of 1, 10, and 50. Consistent with prior research (Zhou et al., 2020; Lu et al., 2021), we report Recall@$k$ (R@$k$) in Section 3.1. However,

| Models | ReR@1 | ReR@10 | ReR@50 | PrR@1 | PrR@10 | PrR@50 | F1@1 | F1@10 | F1@50 | Rec Ratio |
|---|---|---|---|---|---|---|---|---|---|---|
| REDIAL | 0.002 | 0.017 | 0.039 | 0.002 | 0.021 | 0.048 | 0.002 | 0.019 | 0.043 | 0.014 |
| KBRD | 0.008 | 0.040 | 0.096 | 0.011 | 0.052 | 0.126 | 0.009 | 0.045 | 0.109 | 0.317 |
| KGSF | 0.008 | 0.043 | 0.109 | 0.009 | 0.048 | 0.123 | 0.009 | 0.045 | 0.116 | 0.445 |
| REVCORE | 0.006 | 0.031 | 0.077 | 0.014 | 0.075 | 0.183 | 0.008 | 0.044 | 0.109 | 0.203 |
| DialoGPT | 0.011 | 0.070 | 0.172 | 0.011 | 0.071 | 0.174 | 0.011 | 0.070 | 0.173 | 0.462 |
| RecInDial | 0.017 | 0.088 | 0.203 | 0.022 | **0.114** | **0.264** | 0.020 | 0.099 | 0.229 | 0.438 |
| KGSF + ConKD (hard) | **0.023** | 0.110 | 0.249 | **0.024** | 0.113 | 0.257 | **0.023** | 0.111 | **0.253** | 0.499 |
| KGSF + ConKD (soft) | 0.022 | 0.105 | 0.241 | 0.023 | 0.111 | 0.255 | **0.023** | 0.108 | 0.248 | 0.479 |
| DialoGPT + ConKD (hard) | 0.022 | **0.120** | **0.250** | 0.021 | 0.112 | 0.235 | 0.022 | **0.116** | 0.243 | **0.525** |
| DialoGPT + ConKD (soft) | 0.019 | 0.101 | 0.219 | 0.02 | 0.104 | 0.226 | 0.019 | 0.102 | 0.222 | 0.505 |

Table 2: Automatic evaluation results on recommendation task. The scores are averaged over three runs with random seeds. **ReR@**$k$ and **PrR@**$k$ are recall and precision in response for the top $k$ items. **F1@**$k$ is the harmonic mean of **ReR@**$k$ and **PrR@**$k$. **Rec Ratio** is the ratio of recommendation turns to total turns. All scores are computed on final outputs predicted by dialogue modules. Bold and underlined numbers denote the best and second-best performance, respectively.

| Models | Qualitative | | | | Quantitative | | | |
|---|---|---|---|---|---|---|---|---|
| | Flu | Info | Cohe | Average | PPL | DIST-2 | DIST-3 | DIST-4 |
| REDIAL | 1.772 | 0.334 | 1.194 | 1.100 | 17.577 | 0.075 | 0.123 | 0.166 |
| KBRD | 1.640 | 0.729 | 1.321 | 1.230 | 19.039 | 0.123 | 0.223 | 0.304 |
| KGSF | 1.811 | 0.502 | 1.433 | 1.249 | 11.191 | 0.168 | 0.305 | 0.420 |
| REVCORE | 1.854 | 0.512 | 1.187 | 1.184 | 9.283 | 0.098 | 0.170 | 0.235 |
| DialoGPT | 1.766 | 0.781 | 1.580 | 1.376 | 17.552 | **0.206** | **0.419** | **0.595** |
| RecInDial | 1.812 | 0.726 | 1.606 | 1.381 | **5.858** | 0.065 | 0.124 | 0.183 |
| KGSF + ConKD (hard) | 1.831 | 0.856 | 1.571 | 1.419 | 8.886 | 0.138 | 0.249 | 0.344 |
| KGSF + ConKD (soft) | 1.858 | **0.878** | 1.644 | 1.460 | 8.689 | 0.132 | 0.236 | 0.326 |
| DialoGPT + ConKD (hard) | **1.894** | 0.859 | **1.688** | **1.480** | 12.412 | 0.179 | 0.344 | 0.489 |
| DialoGPT + ConKD (soft) | 1.870 | 0.829 | 1.591 | 1.430 | 12.336 | 0.180 | 0.350 | 0.505 |

Table 3: Qualitative and quantitative evaluation results on conversation task. The qualitative scores are averaged over three hired annotators. **Flu**, **Info** and **Cohe** indicate fluency, informativeness, and coherence of model-generated responses. **Average** is the average of the quantities. In quantitative method, scores are averaged over three runs with random seeds. **PPL** is the perplexity of dialogue calculated by a language model, and **DIST**-$n$ is the distinct $n$-gram in corpus level.

given the conversational nature of the task, it is crucial to evaluate recommendation performance *within generated responses*. We introduce Recall in Response (ReR@$k$) following Liang et al. (2021) and Wang et al. (2022a), with a refined calculation approach to ensure scores range within $[0, 1]$. Specifically, we take the average of correct item predictions over the total number of item predictions instead of responses containing items. Additionally, we introduce Precision in Response (PrR@$k$) and compute the harmonic mean of ReR@$k$ and PrR@$k$, denoted as F1@$k$. Furthermore, we assess the system's ability to make active recommendations by introducing the recommendation turn ratio, calculated as the number of dialogue turns with recommended items over the total dialogue turns.

To evaluate dialogue performance, we report perplexity (PPL) and distinct $n$-grams (Li et al., 2016) (DIST), assessing the fluency and the diversity of generated responses, respectively. In prior studies, DIST was computed by counting distinct n-grams

at the corpus-level and averaging them over sentences, which can lead to scores greater than 1. To address this, we have updated the metric calculation to count distinct n-grams and calculate rates at the corpus-level, ensuring the scores fall within the range of 0 to 1. The results evaluated on original metrics are illustrated in Appendix F.

Recent studies find that the $n$-gram based evaluation methods may not be sufficient to assess the performance of a language model (Zhang* et al., 2020). Therefore, we conduct human evaluation to comprehensively assess model performance as done in previous works (Zhou et al., 2020; Wang et al., 2022a). Detailed human evaluation setup is described in Appendix E.

### 4.4 Results

**Evaluation of Recommendation**

In Table 2, we present the recommendation performance of the models. The results clearly demonstrate that models with ConKD (hard) consistently

achieve the highest scores in F1@$k$, indicating superior performance in the recommendation task. Notably, KGSF integrated with ConKD *doubles the scores* compared to KGSF, while DialoGPT with ConKD *achieves the scores 1.5 times as high as* DialoGPT. These improvements are observed not only in single predictions, but also in top-10 and top-50 predictions, indicating superior user-item mapping. We hypothesize that such gain stems from the "dark knowledge" distilled from the recommendation teacher within our framework. This knowledge encompasses inter-class relations that are absent in the one-hot encoded hard targets but are expressed through the probability distribution provided by the recommendation teacher. Furthermore, the models with ConKD make active recommendations, as depicted by the high recommendation ratio, reflecting their focus on task-oriented conversation. Among our gating mechanisms, the hard gate outperforms the soft gate, which can be attributed to the stronger supervision made by the hard gate; a student is guided solely by the recommendation teacher in recommendation time.

**Evaluation of Dialogue Generation**

In addition to the recommendation performances, ConKD exhibits comparable results in conversation metrics, as shown in Table 3. Under quantitative evaluation, the proposed models outperform the backbone models in PPL, indicating enhanced fluency of responses. We observed that RecInDial tends to generate relatively simple responses without active engagement, resulting in lower PPL scores. Regarding the slight decrease in the DIST metric compared to the backbone models in our results, two important observations should be highlighted: 1) The base models fail to effectively address the recommendation task in their responses, and 2) DIST scores alone are insufficient for evaluating the quality of model-generated responses.

These findings are supported by the results of qualitative evaluation, where the single models with ConKD outperform all baselines in average scores. Specifically, when applied to KGSF, the proposed methods significantly enhance informativeness and coherence. This implies our training mechanism performs consistently regardless of the model size, aligning with the automatic evaluation results. We also observe that ConKD-soft outperforms ConKD-hard with KGSF as the backbone, and the opposite holds true for DialoGPT. This discrepancy is attributed to the model capacity of the

| Models | F1@1 | F1@10 | F1@50 |
|---|---|---|---|
| RecInDial | 0.087 | 0.283 | 0.424 |
| DialoGPT + ConKD (hard) | **0.124** | **0.331** | **0.465** |

Table 4: Recommendation results evaluated on contextually relevant items.

dialogue teacher, which influences the gate value. It suggests that the choice between hard and soft gates depends on the capacity of the dialogue teacher.

## 4.5 Quality of Recommendation

In CRS, evaluating recommendations based solely on a single ground-truth item may not fully capture a model's potential, as user preferences can span a wide range of items. To address this, we expand the evaluation by considering contextually relevant items. These relevant items are those located within a 2-hop distance from previously mentioned items in the item-oriented KG (Bizer et al., 2009), sharing attributes such as genre and actor. We compare two models, RecInDial and DialoGPT + ConKD (hard), both of which use the same backbone model to produce unified outputs. Table 4 shows that ConKD consistently outperforms RecInDial across all metrics, with a significant improvement over the single-item evaluation results in Table 2. This highlights ConKD's capability not only to recommend a gold item but also to comprehend the underlying knowledge structures, even in the absence of a knowledge graph.

## 4.6 Efficiency Comparison

Ensuring real-time efficiency is of importance in enhancing the user experience in CRS. This section compares the inference speeds of three models: DialoGPT, DialoGPT + ConKD (hard), and RecInDial. DialoGPT and DialoGPT+ConKD (hard) achieve inference latencies of 5.464ms and 5.306ms per token, respectively. In contrast, RecInDial incurs a slightly higher latency of 6.100ms per token. This additional latency in RecInDial can be attributed to the computation of the knowledge-aware bias and vocabulary pointer. The components involve making decisions between generating items or words in every time step. In ConKD, a language model handles the knowledge-aware recommendations without sacrificing efficiency.

## 4.7 Ablations

To verify the efficacy of each component introduced in this work, we conduct ablation studies

| Models | F1@1 | F1@10 | F1@50 | Rec Ratio |
|--------|------|-------|-------|-----------|
| Vanilla | 0.012 | 0.062 | 0.155 | 0.337 |
| (+) D | 0.017 | 0.063 | 0.165 | 0.334 |
| (+) R | 0.019 | 0.066 | 0.146 | 0.293 |
| (+) D&R | 0.020 | 0.093 | 0.200 | 0.313 |
| (+) D&R&ST | **0.023** | **0.111** | **0.253** | **0.499** |
| $\lambda_t \leftrightarrow 0.5$ | 0.021 | 0.105 | 0.245 | 0.471 |

Table 5: Ablations on the recommendation task. (+) indicates adding corresponding components to Vanilla, a model without ConKD. D and R refer to the dialogue teacher and recommendation teacher. ST is the special token mechanism, and $\leftrightarrow$ indicates replacement. Hard gate is applied for combining the KGSF teachers.

| Models | DIST-1 | DIST-2 | DIST-3 | DIST-4 | PPL |
|--------|--------|--------|--------|--------|-----|
| Vanilla | 0.029 | 0.100 | 0.176 | 0.246 | 7.538 |
| (+) D | 0.029 | 0.100 | 0.172 | 0.241 | 7.071 |
| (+) R | 0.019 | 0.066 | 0.115 | 0.164 | 12.331 |
| (+) D&R | 0.028 | 0.097 | 0.166 | 0.232 | **6.349** |
| (+) D&R&ST | **0.035** | **0.138** | **0.249** | **0.344** | 8.886 |
| $\lambda_t \leftrightarrow 0.5$ | 0.031 | 0.121 | 0.219 | 0.306 | 9.211 |

Table 6: Ablations on the dialogue task.

using several variants of our model. The results are depicted in Table 5 and 6.

We observed that the role of teachers significantly affects the performance of a student. Learning solely from the recommendation teacher enhances recommendation performance but comes at the cost of PPL and DIST scores. In contrast, learning from the dialogue teacher improves fluency. When both teachers are present within the training framework, the student's performance improves in both tasks, highlighting the effectiveness of knowledge transfer through our gating mechanism. Scores decline in all metrics when we replace our gate with a static value of $0.5$. Lastly, the special token also brings meaningful gains, depicted with the increased F1@$k$ and DIST scores.

### 4.8 Case Study

The responses generated by the baselines and ConKD are shown in Table 7. In the conversation, the user has expressed a clear preference for **action** movies and has previously mentioned actors **"bruce willis"** and **"Tom Cruise"**. However, the three baselines generate chit-chat responses without recommendations. Although KGSF suggests a movie, the response lacks coherence and fluency. DialoGPT generates a coherent response but recommends **"Mr.& Mrs.Smith"**, which does not align with the user's preference; neither **Bruce Willis** nor **Tom Cruise** are associated with it. On the other hand, our models produce apt responses that are

| Case 1 | |
|--------|--|
| **Context** | |
| User | Hi! |
| Agent | How are you? |
| User | I like **action** movies. I like **Bruce Willis**. He is a great actor. Or **Tom Cruise**. I am doing well, thanks for asking. |
| **Model Output** | |
| ReDial | What can I help you find today ? |
| KBRD | What would you like? |
| KGSF | I'm doing well. **Edge of Tomorrow** |
| RevCore | I am not sure, but I will have to give you some good ideas. |
| DialoGPT | Have you seen the movie **Mr.& Mrs.Smith**? It is a really good movie...! |
| KGSF* | Have you seen **Edge of Tomorrow**? |
| KGSF** | I like action movies too. Have you seen **Edge of Tomorrow**? |
| DialoGPT* | Have you seen **Mission: Impossible**? |
| DialoGPT** | Have you seen **Mission: Impossible**? |
| Ground Truth | Have you seen **Gladiator**? |

Table 7: Sample outputs generated by the baselines and ConKD. * and ** indicate a model with ConKD (hard) and ConKD (soft) respectively. The last three dialogue turns are depicted for context. Bold words indicate user's preference and recommended items.

both fluent and informative; it makes user-specific recommendations, suggesting **Edge of Tomorrow** and **Mission: Impossible**, both of which are **action** movies featuring **Tom Cruise**. Notably, these recommendations are more aligned with the user's expressed preferences compared to the ground-truth movie **"Gladiator"**, which is an **action** movie without the mentioned actors. Additional examples are provided in the Appendix G.

## 5 Conclusion

In this study, we introduce contextualized knowledge distillation with hard gate and soft gate. In hard gate, a student is either learned from a recommendation teacher or from a dialogue teacher with a discrete value, while the soft gate fuses knowledge from both teachers, removing a hyperparameter from the hard gate. The gates are computed in a context-specific manner by aggregating the probability mass on the interest set (a movie item set in our experiment). Our work verifies the idea in the popular benchmark dataset and the result illustrates the superior performance of our approach compared to strong baselines. In addition, human evaluation mainly conforms with the automatic evaluation, demonstrating that the proposed approach is a well-balanced model with both recommendation and chit-chat ability.

## Limitations

This work is grounded on the student-teacher framework, hence requiring additional computation when obtaining knowledge of a teacher; our approach requires two teachers, one for dialogue and one for recommendation. This extra computation can be a burden in an environment lack of resource. Nonetheless, the proposed approach utilizes a single model for inference. Furthermore, our approach requires the teachers to be well-trained. This, however, is also a shared problem within KD training.

## Ethical Consideration

Since REDIAL dataset (Li et al., 2018) contains multi-turn dialogue histories, the dataset, by nature, may pose a privacy issue. If a dialogue teacher in our framework learns such information, the student in our framework can also learn to output private information while in conversation. Such issue may possibly be handled by employing a privacy classifier model, where a model is trained to identify certain outputs containing private information of a user.

## Acknowledgements

Lei Chen's work is partially supported by National Science Foundation of China (NSFC) under Grant No. U22B2060, the Hong Kong RGC GRF Project 16213620, CRF Project C2004-21GF, RIF Project R6020-19, AOE Project AoE/E-603/18, Theme-based project TRS T41-603/20R, China NSFC No. 61729201, Guangdong Basic and Applied Basic Research Foundation 2019B151530001, Hong Kong ITC ITF grants MHX/078/21 and PRP/004/22FX, Microsoft Research Asia Collaborative Research Grant and HKUST-Webank joint research lab grants.

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

## A  Implementation Details

**Combinations of KGSF and ConKD.** We mainly follow KGSF (Zhou et al., 2020) for the basic model structure of the two teacher and student models. For the dialogue teacher and student, we employ an encoder-decoder structure, with each module consisting of 2 transformer layers. The hidden dimension size of both models is 300. Our work excludes the copy mechanism used in KGSF. In the recommendation model, we utilize 1-layer GNN networks trained with word-oriented and item-oriented KGs as inputs. The normalization constant $Z_{e,r}$ is set to 1. The token classifier is a transformer-based encoder with a classification head. We employ the Adam optimizer (Kingma and Ba, 2014) and apply gradient clipping for stable training. Our chosen hyper-parameters include a batch size of 32, a learning rate of 1e-3, and training for 200 epochs. Furthermore, for ConKD, we set the $\eta$ and $\gamma$ to 0.3 and 0.6, respectively.

**Combinations of DialoGPT and ConKD.** We utilize DialoGPT-small (Zhang et al., 2020) as the backbone for the dialogue teacher and student model. DialoGPT consists of 12 transformer layers and the hidden dimension size of the model is

768. The recommendation teacher and token classifier in the DialoGPT+ConKD models follow the same settings as those in the KGSF+ConKD models. For hyper-parameters, we use a batch size of 32, a learning rate of 1e-3, and train for 20 epochs. For ConKD, we set the $\eta$ and $\gamma$ to 0.6 and 0.4, respectively.

During inference, we only use the student model for the response generation in both settings.

## B    Graph Convolutional Network (GCN)

GCN is adopted to encode the word-oriented KG ConceptNet (Speer et al., 2016). A triple in the KG is denoted by $(w_1, r, w_2)$, where $w_1$ and $w_2$ are word nodes, and $r$ is a relationship between the nodes. The node features are updated with an aggregation function as follows:

$$\mathbf{H}^{(l)} = \mathsf{ReLU}(\tilde{\mathbf{D}}^{-\frac{1}{2}} \tilde{\mathbf{A}} \tilde{\mathbf{D}}^{-\frac{1}{2}} \mathbf{H}^{(l-1)} \mathbf{W}^{(l)}) \quad (13)$$

$\mathbf{H}^{(l)} \in \mathbb{R}^{n \times d}$ and $\mathbf{W}^{(l)}$ denote the node representations and a learnable matrix at the $l$-th layer respectively. $n$ is the number of nodes and $d$ denotes the dimension size of node features. $\tilde{\mathbf{A}} = \mathbf{A} + \mathbf{I}$ is the adjacency matrix of the graph with self-loop, where $\mathbf{I}$ is the identity matrix. $\tilde{\mathbf{D}} = \sum_j \tilde{\mathbf{A}}_{ij}$ refers to a degree matrix.

## C    Validity of $\lambda$

To explore the validity of the $\lambda$, we illustrate variations of the response $y1$ introduced in 3.4.

> $y2$: *If you like romance movies, I would recommend **you** Titanic.*
> $y3$: *If you like romance movies, I would recommend **some** romantic comedies.*

After the word *recommend*, the vocab (*you* and *some*) can be replaced with the item *Titanic* in the $y1$. Therefore, the average value $\lambda_r$ of 0.653 is acceptable in soft gate, indicating *the mass reflects the level to which recommendation is expected.*

## D    Item Refilling

We discuss a two-step inference, in which a response is first generated, and items in the response are refilled with the output of a recommendation module. The recommendation module predicts a probability distribution in each dialogue turn, which remains fixed at each time step during the response generation. Hence, the module *cannot directly handle the variable number of items in a*

*response*. Additionally, the output *does not reflect the dynamic changes of the context*, which fails to provide a context-aware recommendation as seen in the following responses.

> $y4$: ***Blended** with **Adam Sandler** is a fav of mine.*
>
> $y5$: ***Love Actually** with **Adam Sandler** is a fav of mine.*

$y4$ and $y5$ are generated under the original setting and the two-step inference setting respectively[4]. The dialogue module generates the coherent and informative response $y4$, describing the actor **Adam Sandler** who stars in the movie **Blended**. On the other hand, the item refilling fails to incorporate a suitable item into the response, which causes a factual error; there is no relationship between **Adam Sandler** and the movie **Love Actually**. This indicates that the simple integration cannot be the fundamental solution for the mismatch issues; it leads to semantic discrepancy between the recommendations and generated responses. In our unified system, the output distribution over items dynamically changes depending on the context, generating coherent and user-specific responses. This is done by a single model in a single step, thereby reducing the model size and inference time.

## E    Human Evaluation Setup

We engaged three annotators who were tasked with assessing model outputs given a dialogue history. 100 model outputs from each model are randomly sampled and collected, the total being 1000 dialogue turns. The annotators score each model output from the range of 0 to 2 on the level of informativeness, coherence, and fluency of model output. The following instructions were provided to guide annotators in their assessments:

**Fluency:** Fluency encapsulates the naturalness of the generated text. It involves an assessment of how the output adheres to linguistic standards, avoiding grammatical flaws. Annotators should evaluate the syntactic flow, word choice, and overall readability. The scores should be shown as 0, 1, and 2, where each indicates "not fluent", "readable but with some flaws", and "fluent", respectively.

**Informativeness:** The informativeness encompasses the model's ability to convey relevant and

---

[4]The samples are generated by KGSF. We leverage top $k$ sampling to handle the variable number of items in responses, where $k$ is set to 1.

accurate information. Annotators should assess the depth and accuracy of the conveyed information. The scores need to be displayed 0, 1, and 2 where each corresponds to "information is missing or incorrect", "information is included but insufficient or partially inaccurate", and "comprehensive and accurate information", respectively.

**Coherence:** Coherence entails the harmonious integration of the model's output within the evolving conversation. Annotators should assess how well the model comprehends and adheres to the conversation's theme, avoiding abrupt shifts and ensuring a natural conversational flow. The scores should be valued using 0, 1, and 2. Each rating represents "awkward conversation flow", "make sense but somewhat disconnected", and "coherent", respectively.

## F  Results Evaluated Using Original Metrics

| Models | DIST-2 | DIST-3 | DIST-4 | ReR@1 | ReR@10 | ReR@50 |
|---|---|---|---|---|---|---|
| RecInDial | 0.413 | 0.663 | 0.815 | 0.023 | 0.118 | 0.273 |
| Ours | **1.326** | **2.217** | **2.704** | **0.03** | **0.161** | **0.337** |

Table 8: Dialogue and recommendation performance evaluated on the conventional metric. DialoGPT + ConKD (hard) is employed for ours.

## G  Additional Cases

Our models generate diverse movies that differ from the ground truth but align with the user's preferences. For example, in Case 2, when the user requests **old classics**, our models suggest **Gone with the Wind (1939)**, **It's a Wonderful Life (1946)**, **The Big Lebowski (1998)**, **The Outsiders (1967)** and **Driving Miss Daisy (1989)**, all of which are considered old classics. In contrast, other baselines fail to provide recommendation, except for KBRD. In the Case 3, when the user expresses a preference for **family friendly** movies like **Peter Rabbit**, and **Finding Dory**, our models recommend **Beauty and the Beast**, **Jumanji**, **Coco**, and **Troll** all of which are family-friendly, with three of them being animations. This contrasts with other baselines that produce contextually incorrect responses without recommendations or mention **Peter Rabbit** again, which the user had previously mentioned in the dialogue context.

| Case 2 | |
|---|---|
| **Context** | |
| Agent | Hey! What is a movie you love? |
| User | Hello, I am looking for a **old classic**. What would you suggest? |
| **Model Output** | |
| ReDial | Hello |
| KBRD | Have you seen **Jumanji**? |
| KGSF | I can recommend some movies for you? |
| RevCore | I love comedy movies! |
| DialoGPT | Do you have a favorite actor or actress? I love how many types of movies today. |
| KGSF* | Have you seen **Gone with the Wind**? Or maybe something like **It's a Wonderful Life**? |
| KGSF** | Have you seen **The Big Lebowski**? |
| DialoGPT* | I would suggest **The Outsiders It's a classic**! |
| DialoGPT** | Have you seen **Driving Miss Daisy**? |
| Ground Truth | **The Gold Rush** and **Invasion of the Body Snatchers** are classics. |
| **Case 3** | |
| **Context** | |
| User | Hello! Do you know of any **family friendly** movies? |
| Agent | Good Morning have you seen **The Sixth Sense**? |
| User | Yes, but it is **too scary for my children**. something like, **Peter Rabbit** we saw that and it was cute. |
| Agent | A old classic will be **Finding Nemo**. |
| User | Oh yes, and **Finding Dory**. |
| **Model Output** | |
| ReDial | I liked it a lot. |
| KBRD | Glad to help. Have a great day! |
| KGSF | I also recommend watching **Captain Underpants: The First Epic Movie** again. It's pretty good I can think of some people haven't seen |
| RevCore | I think you would like it . |
| DialoGPT | I have not seen **Peter Rabbit** but I have heard good things about it |
| KGSF* | I also really liked **Beauty and the Beast** |
| KGSF** | I also recommend watching the new **Jumanji**. It was cute. |
| DialoGPT* | **Coco** was good too |
| DialoGPT** | I also recommend **Troll**. If your looking at animals that one is really good too! |
| Ground Truth | I saw **Peter Rabbit** with my daughter she loved it" |

Table 9: Additional sample outputs generated by the baselines and ConKD. * and ** indicate a model with ConKD (hard) and ConKD (soft) respectively. The last four dialogue turns are depicted for context. Bold words indicate user's preference and recommended items.