# OpenReview forum: "Towards a Unified Conversational Recommendation System: Multi-task Learning via Contextualized Knowledge Distillation"
_EMNLP/2023/Conference — EMNLP 2023 Main_

### Official Review · Reviewer_mGM1 · 2023-07-28

**Soundness:** 3

**Excitement:**

3: Ambivalent: It has merits (e.g., it reports state-of-the-art results, the idea is nice), but there are key weaknesses (e.g., it describes incremental work), and it can significantly benefit from another round of revision. However, I won't object to accepting it if my co-reviewers champion it.

**Missing References:**

1. Kindly provide methodology comparison to a similar work - Xiaolei Wang, et at., "Towards Unified Conversational Recommender Systems via
Knowledge-Enhanced Prompt Learning"

**Paper Topic And Main Contributions:**

The authors present a novel approach proposing a multi-task learning method for a unified conversational recommendation system. Unlike prior research that employs separate modules for recommendation and dialogue tasks, the proposed approach jointly learns both tasks for recommendation and dialogue semantics respectively. This work also introduces two versions of contextualized knowledge distillation to integrate knowledge from task-specific teachers, which represents a significant contribution to the field. Furthermore, the paper addresses the crucial issue of the discrepancy between recommendation and dialogue tasks in conversational agents, a facet that has not been explicitly explored in previous works. The experimental results provided in the paper demonstrate the efficacy of the proposed approach, showcasing substantial improvements in recommendation performance and dialogue fluency when compared to existing methodologies.

**Questions For The Authors:**

NOTE: ALL my questions were lifted by authors' responses. I do not have further questions.

1. Please elaborate on how the authors compare between their methodology and Xiaolei Wang et al.'s work titled "Towards Unified Conversational Recommender Systems via Knowledge-Enhanced Prompt Learning." Specifically, emphasize the distinct contributions of the proposed approach over a more direct and explainable knowledge graph-based method.

2. Upon thorough examination of the dataset, a noticeable semantic linkage between the prompt and the recommendation, represented by attributes, becomes evident. For instance, instances where the prompt "romantic" corresponds to the recommendation "Titanic." In light of this observation, please justify the advantages of the proposed approach in contrast to semantic-aware recommendation techniques, underscoring its appropriateness and value in the field.

3. Please provide valuable insights into how this research could extend its contributions to other domains of recommendation, such as E-commerce, wherein customers' prompts may lack sufficient informativeness. Consider scenarios such as search queries encountered on platforms like Amazon and eBay.

**Reasons To Accept:**

This article offers a well-structured and motivated research topic, offering clear and novel insights through comprehensible figures and detailed explanations of experimental setups and methodologies, thereby enhancing clarity and reproducibility.

The proposed teacher-student model structure for multi-task learning in a unified conversational recommendation system, which simultaneously addresses recommendation and dialogue tasks, outperforms existing methods in terms of recommendation performance and response fluency, while achieving comparable results in response diversity. This approach demonstrates its unique novelty. The authors rigorously validate the effectiveness of their approach through extensive experiments, along with a detailed ablation study that convincingly justifies the efficacy of the proposed teacher model. Lastly, the qualitative results remain intuitive.

**Reasons To Reject:**

NOTE: The information below were lifted by authors' justification. I do not see such concerns anymore.

The proposed model architecture shows superior performance compared to existing solutions; however, empirical validation of its real-time application efficiency is lacking. For a dialogue-based recommender system, assessing feasibility in real-time application is crucial. To address this, conducting benchmarking against other solutions to evaluate its computational efficiency is recommended. This foundational step will serve as a guide for future research directions.

Furthermore, it is noteworthy that a similar work titled "Towards Unified Conversational Recommender Systems via Knowledge-Enhanced Prompt Learning" exists, which utilizes knowledge graphs instead of introducing additional learnable modules. This approach also demonstrates promising performance, but its methodology for incorporating knowledge priors requires further evaluation and comparison. Although the ablation study shows the effectiveness of the introduced method for learning contextual knowledge, additional evidence is necessary to establish its appropriateness as the optimal approach.

Finally, questions persist regarding the significant contributions of this approach to the field. Existing similar works have evaluated on the same dataset, limited to the domain of movie recommendations. It is essential to demonstrate the generalization ability of this approach to other domains, such as E-commerce recommendation, to establish its broader applicability and impact.

**Reproducibility:**

4: Could mostly reproduce the results, but there may be some variation because of sample variance or minor variations in their interpretation of the protocol or method.

**Reviewer Confidence:**

4: Quite sure. I tried to check the important points carefully. It's unlikely, though conceivable, that I missed something that should affect my ratings.

---

> ### Author Rebuttal · Authors · 2023-08-29
>
> Thank you for your valuable feedbacks and acknowledging the novelty of our approach. We have taken each of your concerns into careful consideration.
>
> **1.	Include empirical validation of real-time application efficiency & comparison with incorporating knowledge graph (KG) approach such as [1]**
>
> We appreciate your emphasis on efficiency evaluation and comparing our approach to KG-incorporating method models such as [1]. Despite our efforts to experiment with the source code provided by [1], we encountered substantial challenges during this process; mainly stemming from the extensive training time required and encountered errors within their codebase. The process involves sequential stages, encompassing semantic fusion and pre-training, conversation prompt training, and recommendation prompt training, which cannot be executed in parallel.
>
> While we remain committed to replicating their work during the remaining rebuttal period, we have also **introduced an alternative baseline, RecInDial [2]**. RecInDial closely aligns with the points raised in your suggestion as well as our research direction. To be specific, a knowledge-aware bias learned from a knowledge graph is incorporated into the language model. Additionally, [1] employs a vocabulary pointer to control predicting either items or words during response generation.
>
> **Performance comparisons**
>
> For a fair comparison, we have chosen to compare DialoGPT + ConKD (hard) with RecInDial in this response, given that DialoGPT-small serves as backbone model for RecInDial.
>
> |          Models         |   DIST-2  |   DIST-3  |   DIST-4  |    PPL    |    F1@1   |   F1@10   |    F@50   | Rec Ratio |
> |:-----------------------:|:---------:|:---------:|:---------:|:---------:|:---------:|:---------:|:---------:|:---------:|
> |        RecInDial        |   0.065   |   0.124   |   0.183   | **5.858** |   0.020   |   0.099   |   0.229   |   0.438   |
> | DialoGPT + ConKD (hard) | **0.179** | **0.344** | **0.489** |   12.412  | **0.022** | **0.116** | **0.243** | **0.525** |
>
> Both models were finetuned on the Redial dataset, and the reported scores are averaged over three runs.
> The results of ConKD are from the current version of our paper, where F1@k represents the harmonic mean of the ReR@k and PrR@k.
>
> In terms of the recommendation metrics (F1@k), it is evident that the results of RecInDial are inferior to ours, *with a notably lower recommendation ratio.* We hypothesize that our contextualized knowledge distillation concept enables the student model to learn class-wise relations which are not addressed by the addition of knowledge bias and vocabulary switching [2].
>
> Furthermore, our results clearly show that the simple integration with vocabulary pointer [2] fails to generate diverse responses, as indicated by the DIST-n scores. We have observed that RecInDial tends to generate relatively simple and concise responses, leading to lower PPL score. While these responses exhibit fluency, they lack the desired level of engagement, as indicated by the low recommendation ratio.
>
> **Efficiency Considerations**
>
> In terms of efficiency, we compared the inference speeds of unified models capable of handling the recommendation and generation tasks in the final outputs. On the Redial dataset, DialoGPT and DialoGPT+ConKD (hard) achieved inference latencies of 5.464ms and 5.306ms per token, respectively. In contrast, RecInDial incurred a slightly higher latency of 6.100ms per token. This additional latency in RecInDial can be attributed to the **computation of the knowledge-aware bias** and **vocabulary pointer**, which controls the choice between generating items or general words **in every time step**. Importantly, our adaptive gates empower a language model to acquire such capability during training, *eliminating the need for additional components in inference.*
>
> In addition, maintaining a knowledge graph for building a bias during inference introduces scalability challenges; the graph size grows with the addition of items and attributes. This is in contrast to our approach, where **a single language model handles the knowledge-aware recommendations.**
>
> In conclusion, our approach outperforms the KG-incorporating baseline [2] and shows superior results in recommendation performances without sacrificing efficiency. We will add the results in the final version of our paper.
>
> **2.	What are the distinct contributions of this approach compared to [1]? Specifically, emphasize the contributions over a more direct and explainable KG-based method.**
>
> The distinct contribution of our method mainly lies in “flexibility” and “efficiency”. Our work is not limited to specific model structures or input formats, allowing flexible adaptation of large language model and recommendation model.
>
> In contrast, [1] introduces a pre-training stage for semantic fusion between word and item embeddings, and the resulted representations are leveraged in conversation and recommendation prompt learning. During inference, they generate a response template first, and then generate a recommendation, which requires to run the language model twice. These sequential procedures during both training and inference not only hinder efficiency but also limit the flexibility of integrating two architectures.
>
>  Furthermore, maintaining a knowledge graph introduces scalability issues as discussed in the answer of question 1. Since the knowledge information is transferred during training, our mechanism eliminates the need for additional components such as vocabulary pointer with knowledge-aware bias [2], and avoids the complexity of a two-step inference [1].
>
>
> **3. What are the advantages of the proposed method over the semantic-aware recommendation (prompting “romantic” may lead to recommend a romance movie)?**
>
> We would like to highlight that the Conversational Recommendation System (CRS) is not merely “controlled text generation”, but a recommendation task within multi-turn dialogues.
>
> For example, not only the top-1 recommendation but also the top-k items are crucial aspect to evaluate the recommendation performance. ConKD enables a single model to learn class-wise relationships (relationship between items) given a dialogue context, e.g., “User: I like romance movie and Leonardo dicaprio” -> “System: I would recommend a romance movie, ***“Titanic:0.8, Lalaland: 0.1, About time: 0.1”***. These relations are expressed through the probability distribution provided by the recommendation teacher, which are not addressed in semantic-aware recommendation.
>
> In addition, when user preferences are represented by multiple attributes or not explicitly stated in a multi-turn dialogue, it is difficult to select relevant attributes to be prompted. ConKD eliminates the need for attribute extraction, enabling contextualized recommendations in multi-attribute or no-attribute scenarios.
>
> **4. How this research could extend its contributions to other domains of recommendation (E-commerce). Consider scenarios such as search queries encountered on Amazon.**
>
>  Our current approach **incorporates knowledge** from the recommendation and dialogue teachers using **adaptive gates**. In e-commerce scenarios, we can adapt this approach in the following ways.
>
> **Explicit and implicit search queries require different knowledge:**
>
> The gate computation can be performed by comparing the information contained in the dialogue teacher and recommendation teacher, taking into account the models’ uncertainty. Specifically, when the user query is vague (resulting in higher uncertainty of the recommendation model due to the absence of specified attributes), the language model can take a more prominent role by providing context-aware recommendations. Conversely, when users provide rich attributes (resulting in lower uncertainty of the recommendation model), the recommendation model can be more engaged to provide an accurate recommendation.
>
> **Handling multiple domains in a contextual manner:**
>
>  Moreover, the gate values can be utilized to effectively manage multiple domains in the e-commerce context.
> For instance, consider a situation where there are distinct recommendation teachers for different product domains. The language model produces contextualized item probabilities given the user query, which can be used as weights for each recommendation teacher from various domains, ensuring that the most relevant suggestions are provided to the user based on the context and preferences.
>
> Thank you for the constructive feedbacks and suggestions. Currently, the public datasets are centered around movie recommendation, but we recognize that e-commerce datasets will be helpful to validate the generalization ability of our model. We will actively explore multi-domain e-commerce datasets during the rebuttal period.
>
>
>
> [1] Towards Unified Conversational Recommender Systems via Knowledge-Enhanced Prompt Learning, KDD 2022
>
> [2] Recindial: A unified framework for conversational recommendation with pretrained language models, AACL 2022

---

### Official Review · Reviewer_nTHZ · 2023-08-05

**Soundness:** 3

**Excitement:**

3: Ambivalent: It has merits (e.g., it reports state-of-the-art results, the idea is nice), but there are key weaknesses (e.g., it describes incremental work), and it can significantly benefit from another round of revision. However, I won't object to accepting it if my co-reviewers champion it.

**Missing References:**

[1] Recindial: A unified framework for conversational recommendation with pretrained language models, AACL 2022

[2] C²-CRS: Coarse-to-Fine Contrastive Learning for Conversational Recommender System, WSDM 22

[3] A Unified Multi-task Learning Framework for Multi-goal Conversational Recommender Systems, TOIS 2023

[4] Towards Unified Conversational Recommender Systems via Knowledge-Enhanced Prompt Learning, KDD 2022

**Paper Topic And Main Contributions:**

This paper proposes a new method for conversational recommendation, which integrates the recommendation and generation with gated knowledge distillation.

**Reasons To Accept:**

The knowledge distillation is first adopted to CRS.

**Reasons To Reject:**

The main issue with this problem is that the authors ignore a crucial and related baseline [1]. To my knowledge, the work titled "RecInDial" [1] delves into the concept of a vocabulary pointer, integrating recommendation and generation modules, and proposing an end-to-end evaluation. Both this work and [1] conduct experiments on the ReDial dataset and evaluate the recommendation performance in an end-to-end manner, with [1] demonstrating significantly better performance than this work.

In addition to [1], which presents the most analogous idea and evaluation framework, there exist other recent publications in the field of Conversational Recommender Systems that have not been compared or reviewed, namely [2], [3], and [4].

Here are the references for the mentioned publications:

[1] Recindial: A unified framework for conversational recommendation with pretrained language models, AACL 2022

[2] C²-CRS: Coarse-to-Fine Contrastive Learning for Conversational Recommender System, WSDM 2022

[3] A Unified Multi-task Learning Framework for Multi-goal Conversational Recommender Systems, TOIS 2023

[4] Towards Unified Conversational Recommender Systems via Knowledge-Enhanced Prompt Learning, KDD 2022

Although I appreciate the integration of knowledge distillation and gated switching mechanisms in this work, the rationale behind incorporating these mechanisms remains insufficiently explained. Additionally, the performance improvements resulting from the proposed method are not significant.

What’s more, the experimental results reported in this paper is not convincing.

1. the evaluation results in the orginal papers (KGSF, KBRD, RevCore) is much higher than the results listed in this paper, the author did not give a clear explanation.

2.the evaluations of generation are not consistent in Qualitative and Quantitive results, see Table 3. For example, KGSF + KonKD(hard) shows generally the best performance in Qualitative results, but it shows almost the worst among the variants.

3.The human evaluation lacks credibility due to the inadequately described valuation scheme.


**Reproducibility:**

3: Could reproduce the results with some difficulty. The settings of parameters are underspecified or subjectively determined; the training/evaluation data are not widely available.

**Reviewer Confidence:**

4: Quite sure. I tried to check the important points carefully. It's unlikely, though conceivable, that I missed something that should affect my ratings.

---

> ### Author Rebuttal · Authors · 2023-08-29
>
> We appreciate your constructive feedback and comments. We have taken each of your concerns and questions into careful consideration.
>
> **1. Comparison to “RecInDial [1]” which performs significantly better than this work.**
>
> Thank you for pointing out the comparison with RecInDial [1]. We would like to highlight two key aspects: (1) **The marked variations in performance** between our work and [1] can be attributed to **changes we made in metrics calculation** (2) We have implemented [1], and **the results confirm the first point**.
>
> **(1) Changes in metrics calculation**
>
> - DIST-n in Table 3
>
>     In Table 3, the baselines utilize the KBRD approach for computing the DIST-n scores. This methodology involves counting the distinct n-grams at the **corpus-level** and then averaging them over **sentences**. Such method can lead to scores greater than 1, as mentioned in the following GitHub issue ([https://github.com/neural-dialogue-metrics/Distinct-N/issues/2](https://github.com/neural-dialogue-metrics/Distinct-N/issues/2)). To address this, we have updated our evaluation code to count distinct n-grams and **calculate rates at the corpus-level**, ensuring the scores fall within the range of 0 to 1.
>
> - ReR@k in Table 2
>
>     In the context of ReR@k (Recall in Response) metrics shown in Table 2, we counted the correct movies and averaging them in a token-level. The reported results in [1] were replicated by counting the correct movies in a token-level and averaging them in a sentence-level.
>
> **(2) Experiment results**
>
> - Performance comparison using different metric calculation methods
>
>     Considering that DialoGPT-small serves as a backbone model for RecInDial, we have compared DialoGPT + ConKD (hard) to the baseline for a fair assessment. Both models were finetuned on the Redial dataset, and the reported scores are averaged over three runs. The results are presented in the tables below:
>
>     **[Baseline metrics calculation method]**
>
>     |          Models         |   Dist-2  |   Dist-3  |   Dist-4  |   ReR@1  |   ReR@10  |   ReR@50  |
>     |:-----------------------:|:---------:|:---------:|:---------:|:--------:|:---------:|:---------:|
>     |        RecInDial        |   0.413   |   0.663   |   0.815   |   0.023  |   0.118   |   0.273   |
>     | DialoGPT + ConKD (hard) | **1.326** | **2.217** | **2.704** | **0.03** | **0.161** | **0.337** |
>
>     **[Our metrics calculation method]**
>
>     |          Models         |   Dist-2  |   Dist-3  |   Dist-4  |   ReR@1   |  ReR@10  |  ReR@50  |
>     |:-----------------------:|:---------:|:---------:|:---------:|:---------:|:--------:|:--------:|
>     |        RecInDial        |   0.065   |   0.124   |   0.183   |   0.017   |   0.088  |   0.203  |
>     | DialoGPT + ConKD (hard) | **0.179** | **0.344** | **0.489** | **0.022** | **0.12** | **0.25** |
>
>     In both methods used for metric calculations, it shows that our approach consistently outperforms RecInDial. In terms of the recommendation metrics (ReR@k), despite having a gain over DialoGPT as shown in Table 2, the results of RecInDial are inferior to those of ours. We hypothesize that *our contextualized knowledge distillation concept enables the student model to learn class-wise relations, which are not addressed by the addition of knowledge bias and vocabulary switching [1]*. Furthermore, it is evident that the simple integration with vocabulary pointer [1] fails to generate diverse responses, as indicated by the DIST-n scores.
>
> - Overall performances
>
>     We provide overall performances with additional evaluation metrics outlined in our paper, including DIST-n and PPL for dialogue performance, as well as the F1@k (harmonic mean of ReR@k and PrR@k) and Rec Ratio for recommendation performance.
>
>     |          Models         |   DIST-2  |   DIST-3  |   DIST-4  |    PPL    |    F1@1   |   F1@10   |    F@50   | Rec Ratio |
>     |:-----------------------:|:---------:|:---------:|:---------:|:---------:|:---------:|:---------:|:---------:|:---------:|
>     |        RecInDial        |   0.065   |   0.124   |   0.183   | **5.858** |   0.020   |   0.099   |   0.229   |   0.438   |
>     | DialoGPT + ConKD (hard) | **0.179** | **0.344** | **0.489** |   12.412  | **0.022** | **0.116** | **0.243** | **0.525** |
>
>     We have observed that RecInDial tends to generate relatively simple and concise responses, resulting in lower PPL scores. While these responses exhibit fluency, they lack the desired level of engagement, as indicated by the low recommendation ratio.
>
> - Efficiency considerations
>
>     In terms of efficiency, we compared the inference speeds of unified models capable of handling the recommendation and generation tasks in the final outputs.  On the Redial dataset, DialoGPT and DialoGPT+ConKD (hard) achieved inference latencies of 5.464ms and 5.306ms per token, respectively. In contrast, RecInDial incurred a slightly higher latency of 6.100ms per token. This additional latency in RecInDial can be attributed to the **computation of the knowledge-aware bias** and **vocabulary pointer**, which controls the choice between generating items or general words **in every time step**. Importantly, *our adaptive gates empower a language model to acquire such capability during training, eliminating the need for additional components in inference*.
>
>     In addition, maintaining a knowledge graph for building user representations during inference introduces scalability challenges; the graph size grows with the addition of items and attributes. This is in contrast to our approach, where *a single language model handles the knowledge-aware recommendations.*
>
>     In conclusion, our approach outperforms the baseline and shows superior results in recommendation performances without sacrificing efficiency. We will add the results and changes in metric calculations in the final version of our paper.
>
> **2. Other recent baselines**
>
> We appreciate your mention of other recent baselines, and we have carefully reviewed the ones you referenced. Upon our review, we observed that [2] is particularly well-aligned with our research direction and has achieved the promising performance results.
> Despite our efforts to experiment with the source code provided by [2], we encountered substantial challenges during this process; mainly stemming from the extensive training time required and encountered errors within their codebase.  The process involves sequential stages, encompassing semantic fusion and pre-training, conversation prompt training and recommendation prompt training, which cannot be executed in parallel. These factors made it difficult for us to successfully reproduce their results within the limited time.
>
> We are fully committed to replicating their work during the remaining rebuttal period.
>
> **3.	Lack of explanation regarding the rationale behind the integration of knowledge distillation and gated switching mechanisms.**
>
> Thank you for your feedback and for recognizing the integration of knowledge distillation and gated switching mechanisms in our work. We would like to clarify the rationale behind the integration.
>
> In the context of natural language generation, especially in Conversational Recommendation, different knowledge is required at each time step to complete a proper response. *The core motivation behind incorporating adaptive gate mechanism into knowledge distillation is to address two fundamental questions: **“What knowledge should we distill”** and **“How much knowledge to be distilled”** at this time step.* Within our framework, a dialogue teacher assesses these aspects dynamically at every time step. *This enables us to adaptively measure the quantity of each knowledge to distill in a contextual manner.* This is built on an assumption that the sum of item probabilities mapped by the dialogue teacher reflects the extent to which recommendation is expected. The validation of the assumption is also provided in Line 343-354 of our paper.
>
> Additionally, our mechanism offers the advantage of *extending knowledge transfer beyond the specific movie indexes*. In our approach, the contextualized gates are computed by aggregating the probability mass on the interest set, rather than identifying specific indexes. Thus, the knowledge transfer can be performed across multiple domains in a contextual manner. Furthermore, our mechanism eliminates the need for additional components such as vocabulary pointer with knowledge-aware bias [1], and avoids the complexity of a two-step inference (generate and revise) [2].
>
> We will add this detailed explanation in the final version of our paper.
>
> **4.	The performance improvements resulting from the proposed method are not significant.**
>
> We would like to clarify that, while our models do not always achieve the top rankings in certain dialogue metrics within automatic evaluation results, it's essential to recognize that *our approach achieves a well-balanced combination of the dialogue and recommendation capabilities with significant improvement in recommendation metrics*. In many cases, achieving the best performance in one specific metric can sometimes come at the cost of sacrificing performance in other areas.
>
> **5.	The experimental results reported in this paper is not convincing**
>
> We have clarified the details of evaluation in the answer of the question 1.
>
> **6.	Clarify the inconsistency between qualitative and quantitative evaluation results.**
>
> We appreciate your observation regarding the potential inconsistency between our qualitative and quantitative evaluation results. We would like to clarify the distinction between these two evaluation approaches. The qualitative evaluation assesses more nuanced properties of generated responses, such as informativeness and coherence. On the other hand, quantitative evaluation focuses on assessing response diversity. It’s essential to recognize that *quantitative results can indicate positive performance even if a language model generates responses that are meaningless or lack fluency, as long as the generated tokens are diverse*.
> In terms of the fluency which is used in both evaluations (Flu in Qualitative and PPL in Quantitative) as shown in Table 3, we would like to highlight that our KGSF+ConKD (hard) model consistently performs the best. Furthermore, our KGSF+ConKD (soft) model ranks as the second-best model.
>
> **7.	Provide details of the human evaluation scheme.**
>
> Thank you for pointing out the lack of details in human evaluation scheme. In addition to Appendix G, we provide the detailed guildelines provided to the human annotators below.
>
> Fluency: Fluency encapsulates the naturalness of the generated text. It involves an assessment of how the output adheres to linguistic standards, avoiding grammatical flaws. Annotators should evaluate the syntactic flow, word choice, and overall readability. The scores should be shown as 0, 1, and 2, where each indicates “not fluent”, “readable but with some flaws”, and “fluent”, respectively.
>
> Informativeness: The informativeness encompasses the model's ability to convey relevant and accurate information. Annotators should assess the depth and accuracy of the conveyed information. The scores need to be displayed 0, 1, and 2 where each corresponds to “information is missing or incorrect”, “information is included but insufficient or partially inaccurate”, and “comprehensive and accurate information”, respectively.
>
> Coherence: Coherence entails the harmonious integration of the model's output within the evolving conversation. Annotators should assess how well the model comprehends and adheres to the conversation's theme, avoiding abrupt shifts and ensuring a natural conversational flow. The scores should be valued using 0, 1, and 2. Each rating represents  “awkward conversation flow”,  “make sense but somewhat disconnected”, and “coherent”, respectively.
>
> We will add this detailed description in the final version of our paper.
>
> [1] Recindial: A unified framework for conversational recommendation with pretrained language models, AACL 2022
>
> [2] Towards Unified Conversational Recommender Systems via Knowledge-Enhanced Prompt Learning, KDD 2022

---

### Official Review · Reviewer_hs9f · 2023-08-05

**Soundness:** 3

**Excitement:**

3: Ambivalent: It has merits (e.g., it reports state-of-the-art results, the idea is nice), but there are key weaknesses (e.g., it describes incremental work), and it can significantly benefit from another round of revision. However, I won't object to accepting it if my co-reviewers champion it.

**Paper Topic And Main Contributions:**

Positioning: the paper is about conversational recommender systems - systems that recommend items or entities to users through the course of a dialog. Specifically, the authors note consistency challenges between recommendation and dialog modules in multi-part ConvRec systems. They propose to learn an E2E conversational recommender by distilling recommender systems and dialog models into a single "Unified" model.

**Reasons To Accept:**

- Section 3.1 preliminary experiments are well laid-out and the results justify the challenge tackled by the paper. The discrepancies between recommendations made by the recommender module and those present in the generated dialog for several multi-stage ConvRec systems is substantial (70-90%) and demonstrates a large gap.
- The system design should be clarified - using task-specific signal tokens ([REC]/[GEN]) etc. can all be done with a language model alone, and one could perform multi-task learning with the recommender system or teacher-forcing alone with the recommender system without the dialog system. However, the authors do justify their system design via ablations in 4.3
- The ConKD approach significantly improves response entity precision and recall compared to baselines (Table 2).

**Reasons To Reject:**

- The "comprehensive evaluation" by using KGSF/DialoGPT is a bit of a strange assertion; the effect of pre-trained language models are orthogonal to the effect of semantic fusion with knowledge graph inputs (a la KGSF). It would have been nice to see a knowledge graph fused to DialoGPT or some similar method (e.g. via prompting prepended with a subgraph or other KG features).
- The case study is confusing and not particularly useful; all models evaluated demonstrate a failure to recommend the appropriate item compared to ground truth, and there isn't enough of a discussion to figure out what we're supposed to digest from this section.
- The ReDial dataset is an interesting benchmark but limited to a single domain with a relatively small corpus of items and straightforward user behavior. The paper would be better served by evaluating on a dataset like GoRecDial (Kang 2019), which demonstrates more complex recommendation/querying behavior.

**Reproducibility:**

3: Could reproduce the results with some difficulty. The settings of parameters are underspecified or subjectively determined; the training/evaluation data are not widely available.

**Reviewer Confidence:**

4: Quite sure. I tried to check the important points carefully. It's unlikely, though conceivable, that I missed something that should affect my ratings.

**Typos Grammar Style And Presentation Improvements:**

- The diagram on Page 1 is difficult to read; the font is quite small and it's difficult to parse. Perhaps it could be tidied up.

---

> ### Author Rebuttal · Authors · 2023-08-29
>
> Thank you for your thorough review and recognition of the rationale behind our system design, as well as the effectiveness of our approach. We have given careful thought to all your concerns and questions.
>
> **1. Add a baseline that fuses knowledge graph inputs and pre-trained language model.**
>
> Thank you for your suggestion in adding a baseline. We have implemented RecInDial [1], which closely aligns with the points raised in your suggestion as well as our research direction. [1] integrates knowledge graph inputs and a pre-trained language model by introducing a knowledge-aware bias, computed through the aggregation of node representations. Additionally, [1] employs vocabulary pointer to control when to predict items or words during response generation.
>
> **Performance comparisons**
>
> For a fair comparison, we have chosen to compare DialoGPT + ConKD (hard) with RecInDial in this response, given that DialoGPT-small serves as backbone model for RecInDial. Both models were finetuned on the Redial dataset, and the reported scores are averaged over three runs.
>
> |          Models         |   DIST-2  |   DIST-3  |   DIST-4  |    PPL    |    F1@1   |   F1@10   |    F@50   | Rec Ratio |
> |:-----------------------:|:---------:|:---------:|:---------:|:---------:|:---------:|:---------:|:---------:|:---------:|
> |        RecInDial        |   0.065   |   0.124   |   0.183   | **5.858** |   0.020   |   0.099   |   0.229   |   0.438   |
> | DialoGPT + ConKD (hard) | **0.179** | **0.344** | **0.489** |   12.412  | **0.022** | **0.116** | **0.243** | **0.525** |
>
> The results of ConKD are from the current version of our paper, where F1@k represents the harmonic mean of the ReR@k and PrR@k.
>
> In terms of the recommendation metrics (F1@k), it is evident that the results of RecInDial are inferior to ours, with a notably lower recommendation ratio. We hypothesize that our contextualized knowledge distillation concept enables the student model to learn class-    wise relations which are not addressed by the addition of knowledge bias and vocabulary switching [1].
>
> Furthermore, our results clearly show that the simple integration with vocabulary pointer [1] fails to generate diverse responses, as indicated by the DIST-n scores. We have observed that RecInDial tends to generate relatively simple and concise responses, leading to lower PPL score. While these responses exhibit fluency, they lack the desired level of engagement, as indicated by the low recommendation ratio.
>
> **Efficiency considerations**
>
> In terms of efficiency, we conducted a comparison of inference speeds. On the Redial dataset, DialoGPT and DialoGPT+ConKD (hard) achieved inference latencies of 5.464ms and 5.306ms per token, respectively. In contrast, RecInDial incurred a slightly higher latency of 6.100ms per token. This additional latency in RecInDial can be attributed to the **computation of the knowledge-aware bias** and **vocabulary pointer**, which controls the choice between generating items or general words **in every time step**. Importantly, our adaptive gates empower a language model to acquire such capability during training, eliminating the need for additional components in inference.
>
> In addition, maintaining a knowledge graph for building user representations during inference introduces scalability challenges; the graph size grows with the addition of items and attributes. This is in contrast to our approach, where **a single language model handles the knowledge-aware recommendations.**
>
> In conclusion, our approach outperforms the baseline and shows superior results in recommendation performances without sacrificing efficiency. We will add the results in the final version of our paper.
>
> **2.	Lack of discussion in the Case Study: All models fail to recommend the appropriate item compared to ground truth. What to digest in this section?**
>
> We would like to clarify that evaluating solely based on ground-truth recommendations may not capture the full potential of a conversational recommendation system, as there can be a wide array of items that align with a user's preferences.
>
> For instance, in Table 6, the ground-truth recommended movie **“Gladiator”** aligns only with the user’s preferred genre **“action”**, while the user’s preferred actors **“Bruce Willis”** or **“Tom Cruise”** are not associated with the movie **“Gladiator”**. However, our models recommend movies **“Edge of Tomorrow”** and **“Mission: Impossible”** which are action movies featuring **“Tom Cruise”**, as described in the Line 514-519.
>
> In Appendix H, our models generate diverse movies that differ from the ground truth but align with the user’s preferences. For example, in Case 2, when the user requests **“old classics”**, our models suggest movies **“Gone with the Wind (1939)”**, **“It’s a Wonderful Life (1946)”**, **“The Big Lebowski (1998)**”, **“The Outsiders (1967)”** and **“Driving Miss Daisy (1989)”**, all of which are considered old classics. In contrast, other baselines fail to provide recommendation, except for KBRD.
>
> In the Case 2, when the user expresses a preference for “**family friendly”** movies like **“Peter Rabbit”**, and **“Finding Dory”**, our models recommend **“Beauty and the Beast”, “Jumanji”, “Coco”**, and **“Troll”** all of which are **family-friendly**, with three of them being animations. This contrasts with other baselines that produces contextually incorrect responses without recommendations and the ground truth that includes the movie **“Peter Rabbit”** again, which the user had previously mentioned in the dialogue context. We will include this evaluation approach and further discussion on the Case 2 and Case 3 in Appendix H.
>
> **3.	Add the “GoRecDial” dataset**
>
> Thank you for your helpful comment. We have carefully reviewed the dataset, and while it presents an interesting approach, it primarily focuses on a game-like scenario where the goal is to "find" the correct movie among a set of five options within minimal dialogue turns, with the dialogue concluding once the correct movie is successfully identified. This setup closely aligns with policy-based reinforcement learning methods, which are typically adopted for such tasks, rather than our task, which involves a combination of open-ended chitchat and recommendation. Additionally, the GoRecDial dataset is also centered around a single domain-movie recommendation, and is relatively small in scale compared to the Redial dataset we used.
>
> However, we recognize that additional datasets will be helpful to validate the generalization ability of our model. We will actively explore multi-domain datasets during the rebuttal period.
>
> [1] Recindial: A unified framework for conversational recommendation with pretrained language models, AACL 2022

---

### Meta-Review · Area_Chair_zD6b · 2023-09-20

**Recommendation:** 4

**Metareview:**

This paper proposes a new method for conversational recommendation, which integrates recommendation and generation with gated knowledge distillation.

The authors carefully addressed most concerns (e.g., missing crucial baselines and unconvincing experimental results) from the reviewers in the rebuttal. The authors are encouraged to add all the new experimental results in their paper if accepted. The authors should also make it extremely clear that they have revised the evaluation metric. It would be nice to add results corresponding to the original metric in the appendix as well.

---

### Decision · Program_Chairs · 2023-10-07

**Decision:**

Accept-Main

**Comment:**

This paper proposes a new method for conversational recommendation, which integrates recommendation and generation with gated knowledge distillation.

The authors carefully addressed most concerns (e.g., missing crucial baselines and unconvincing experimental results) from the reviewers in the rebuttal. The authors are encouraged to add all the new experimental results in their paper if accepted. The authors should also make it extremely clear that they have revised the evaluation metric. It would be nice to add results corresponding to the original metric in the appendix as well.